# Prevalence of the Bifid Mandibular Condyle and Its Relationship with Pathologies of the Temporomandibular Joint: A Systematic Review and Meta-Analysis

**DOI:** 10.3390/diagnostics13203282

**Published:** 2023-10-23

**Authors:** Juan José Valenzuela-Fuenzalida, Kora-lle Keller Navarro, Pia Urbina, Martin Trujillo-Riveros, Pablo Nova-Baeza, Mathias Orellana-Donoso, Macarena Rodriguez-Luengo, Alvaro Beccerra Farfan, Juan A. Sanchis-Gimeno

**Affiliations:** 1Department of Morphology and Function, Faculty of Health and Social Sciences, Universidad de Las Américas, Santiago 8370040, Chile; koraa.keller@gmail.com; 2Departamento de Morfología, Facultad de Medicina, Universidad Andrés Bello, Santiago 8370146, Chile; piaurbinatamayo@gmail.com (P.U.); martintrujilloriveros@gmail.com (M.T.-R.); pablo.nova@usach.cl (P.N.-B.); miorellanadonoso@gmail.com (M.O.-D.); macarena.rodriguez@unab.cl (M.R.-L.); 3Escuela de Medicina, Universidad Finis Terrae, Santiago 7501015, Chile; 4Departamento de Ciencias Química y Biológicas, Facultad de Ciencias de la Salud, Universidad Bernardo O’Higgins, Santiago 8370993, Chile; alvaro.becerra@ubo.cl; 5Department of Human Anatomy and Physiology, Faculty of Health Sciences, University of Johannesburg, Johannesburg 2092, South Africa; juan.sanchis@uv.es; 6GIAVAL Research Group, Department of Anatomy and Human Embryology, Faculty of Medicine, University of Valencia, 46001 Valencia, Spain

**Keywords:** anatomy mandibular condyle, variation mandibular condyle, bifid mandibular condyle, trifid mandibular condyle, anatomical variation, clinical anatomy, temporomandibular joint pathologies

## Abstract

Objective: The aim of this study was to describe the prevalence of anatomical variants in the bifid mandibular condyle (BMC) and report its association with temporomandibular joint (TMJ) pathology. Methods: We searched the Medline, Scopus, Web of Science, Google Scholar, CINAHL, and LILACS databases from their inception up to September 2023. Two authors independently performed the search, study selection, and data extraction, and they also assessed the methodological quality with an assurance tool for anatomical studies (AQUA). Finally, the pooled prevalence was estimated using a random effects model. Results: A total of 50 studies met the eligibility criteria. Twenty studies, with a total of 88,625 subjects, were included in the meta-analysis. The overall prevalence of the bifid mandibular condyle (BMC) variant was 1% (95% CI = 1% to 2%). Conclusions: The correlation between the BMC and TMJ pathologies has a relatively low prevalence in studies that present a considerable number of subjects. From a clinical point of view, a direct association cannot be made between the presence of the BMC and TMJ pathologies or symptoms.

## 1. Introduction

The mandible is a bone belonging to the viscerocranium, the only mobile bone of the skull. Structurally, it is made up of two components: the mandibular body and the mandibular ramus, each with different anatomical characteristics and repairs that fulfill fundamental roles, such as providing articulation to the lower teeth, in addition to allowing the passage of neurovascular structures of importance for the oral and dental region [1,2,3]. In the branch of the mandible are the coronoid and condylar processes; the latter acts as an articular component within the mandible, which allows it to articulate with the mandibular fossa of the temporal bone, thus forming the temporomandibular joint (TMJ). A malformation or variant of any articular or bone component of the TMJ will alter the function of the region. One of the variants of the articular bone components is the multi-headed condyle, a term used to define a rare anomaly that affects the condyle [4,5,6]. The mandibular condyle with two heads is called the bifid mandibular condyle (BMC) (Figure 1). This is characterized by the fact that the head of the mandibular condyle is duplicated and can be both articular components or only one of the two bifurcated condyles. On the other hand, the condyle can be divided into three heads, which is known as the trifid mandibular condyle. Some cases have also been discovered with a condyle divided into four heads, called a tetraphid condyle [7]. Demographically, the BMC does not seem to affect individuals of any specific ethnicity, race, sex, or age, but some literature has reported that the majority of cases are between 3 and 67 years of age. The prevalence of the BMC is controversial, since it varies widely between different published studies, with the most common being 0.3% to 1.82% (4–6). The BMC can be asymptomatic or present different signs and symptoms, such as pain, swelling, hypomobility, joint blockage, deviation, joint dislocation, or even TMJ ankylosis. This highlights the importance of understanding this entity to identify its possible causative factors, and to know the types and degrees of joint dysfunction that can arise without clinical intervention and how these could functionally alter the TMJ [8,9,10,11].

The objective of this review is to understand the prevalence and characteristics of the BMC and its relationship with TMJ pathologies.

## 2. Methods

### 2.1. Protocol

This systematic review and meta-analysis were performed and reported according to the Preferred Reporting Items for Systematic Reviews and Meta-Analyses (PRISMA) statement [12].

### 2.2. Eligibility Criteria

Studies on the presence of variants and their association with any clinical condition were considered eligible for inclusion if the following criteria were met: (1) population: availability of dissection specimens or BMC images; (2) results: prevalence of the BMC, variants, and their correlation with TMJ pathologies or surrounding regions; and (3) studies: inclusion of research articles, research reports, or original research published in English or Spanish in peer-reviewed journals and indexed in the reviewed databases (listed in Section 2.3). On the other hand, the exclusion criteria were as follows: (1) population: animal studies; (2) studies that analyzed variants from other regions, such as the mandibular ramus, coronoid process, or other neighboring structures only; and (3) studies published as letters to the editor or comments.

### 2.3. Electronic Search

We systematically searched MEDLINE (via PubMed), Web of Science, Google Scholar, Cumulative Index to Nursing and Allied Health Literature (CINAHL), Scopus, and EMBASE from 1990 to September 2023.

The search strategy included a combination of the following terms: “Anatomy mandibular condyle” (No MeSH), “variation mandibular condyle” (No MeSH), “bifid mandibular condyle ” (No MeSH terms), “trifid mandibular condyle” (No MeSH), “variation anatomical” (No MeSH), “clinical anatomy” (No MeSH), and “pathologies temporomandibular joint” (No MeSH), using the Boolean connectors “AND”, “OR”, and “NOT”. 

### 2.4. Study Selection

Two authors (JJV and MO) independently screened the titles and abstracts of the references retrieved from the search. We obtained the full texts of the references that either author considered potentially relevant. A third reviewer (MR) was included if a consensus could not be reached.

### 2.5. Data Collection Process

Two authors (PU and JJV) independently extracted data on the outcomes of each study. The following data were extracted from the original reports: (1) authors and year of publication; (2) country; (3) type of study; (4) sample characteristics (sample size, age, distribution, and sex); (5) prevalence and morphological characteristics of the BMC; (6) statistical data reported by each study; and (7) laterality of the variant.

### 2.6. Assessment of the Methodological Quality of the Included Studies

Quality assessment of the retrospective and prospective observational studies was performed using the methodological quality assurance for anatomical studies (AQUA) tool proposed by the International Evidence-Based Anatomy Working Group [13]. Two reviewers (JJV and PN) independently performed the data extraction and quality assessment. A third reviewer (KK) was involved if a consensus could not be reached. For case study bias, two authors (JS and MR) separately assessed the risk of bias. To bias the case studies, the Joanna Briggs Institute assessment tool for case reports was used [14].This questionnaire has eight items, with answers such as “yes”, “unclear”, “no”, or “not applicable”, with the following criteria to be evaluated: (1) low risk of bias: more than 70% score of “yes”, (2) moderate risk of bias: 50–69% score of “yes”, and (3) high risk of bias: less than 49% score of “yes”.

### 2.7. Statistical Methods

The data extracted from the meta-analysis were interpreted by calculating the prevalence of BMC variants using the JAMOVI software 2.4.8 version [14]. For the appropriate statistical model for the analysis of the data obtained, we used the DerSimonian–Laird model. Additionally, a random effects model was used because BMC prevalence data were heterogeneous. To calculate heterogeneity, we use the chi-square test (I^2^). For the chi-square test, the *p*-value proposed by the Cochrane Collaboration was considered significant when it was <0.10. The values of the I^2^ statistic were interpreted with a 95% confidence interval [CI] as follows: 0–40% might not be important, 30–60% might indicate moderate heterogeneity, 50–90% might represent substantial heterogeneity, and 75–100% could represent a significant amount of heterogeneity [15]. 

## 3. Results

### 3.1. Included Articles

The search resulted in a total of 1097 articles from different databases that met the criteria and search terms established by the research team. The filter was applied to the titles and/or abstracts of the articles in the consulted databases, and the primary criterion of elimination of duplicates was used. In total, 180 full-text articles were evaluated for eligibility for inclusion in this meta-analysis and systematic review. Next, 148 studies were excluded because their primary and secondary results did not match those of this review or because they did not meet the established criteria for good data extraction, resulting in 32 articles being included for analysis (*n* = 88,625 patients, images, and cadavers) (Figure 2).

### 3.2. Characteristics of the Studies and the Study Population

Among the 32 included studies, 11 were case reports, 21 were retrospective studies and none were prospective studies. The samples included in the reviewed studies were geographically distributed across all continents except Africa and Oceania. A total of 11 studies were conducted in Europe which is equivalent to 34.37%. The cumulative number of patients in these studies was 34,750, accounting for 39.21% of the reviewed samples. Among the reviewed studies, 14 were conducted in Asia, which was equivalent to 43.75% of the studies included in this review; the cumulative number of patients in these studies was 3395, which is equivalent to 3.83% of samples included in our analysis. Three studies were conducted in North America, which is equivalent to 9.37% of the studies included in this review. It should be noted that the cumulative number of patients in these three studies was 24, which accounts for 0.03% of the sample size included in our analysis. Four studies were conducted in South America, accounting for 12.5% of the studies included in this review, and the cumulative number of patients in these studies was 50,456, or 56.93% of the analyzed samples (Table 1 and Figure 3).

Regarding the 32 studies that included the characteristics of laterality of the jaws with the BMC, only the jaws with the variant were included in this analysis, since we believe that including all jaws that were analyzed would overestimate the results and would not be representative of the characteristics of the BMC. Six of the thirty-two included studies did not report the laterality of jaws with anatomical variants; for the studies that showed the laterality of the anatomical variants, 66 jaws presented bilaterality of the BMC, while 248 jaws presented the BMC unilaterally.

For the sex characteristics in the 32 studies included in this review, we will show the sex of the subjects who presented the variant. Sex was not reported in 10,309 jaws with the BMC variant, which is equivalent to 11.63% of all included subjects. In this review, for the studies that presented the sex of the sample with the BMC, 144 were male, equivalent to 0.16%, while 168 were female, equivalent to 0.18%. It should be noted that the results are only expressed for the variant and not for the total sample, since if we included the entire sample of studies, the results could be overestimated. Regarding the detection methods of BMC in the included studies, 13 studies identified BMC through panoramic radiography, 6 studies through CT scan. On the other hand, the largest number of subjects was detected through panoramic radiography (81,086 subjects), then through cone beam computerized tomography (3110 subjects), and finally through macroscopic evaluation (corpse) (2525 subjects) (Table 2).

### 3.3. Prevalence and Risk of Bias

For the meta-analysis of the prevalence of the BMC, 15 studies were included [16,17,22,23,24,25,27,31,32,35,36,37,38,39,47], with a prevalence of 1% with a deviation standard from 1 to 2%, showing a heterogeneity of (I^2^ = 82%) (Table 3 and Figure 4). Regarding the risk of bias in the case studies, in a total of 11 studies, 100% of the articles analyzed presented a low risk of bias. If we analyze the different items one by one, only question eight remains. “Does the case report provide takeaway lessons?” presented a high risk of bias in eight of the eleven articles analyzed [18,19,20,21,28,29,33,34,40,41,45] (Table 4 and Figure 5). For the analysis of biases using AQUA, 22 studies were included [16,17,22,23,24,25,26,27,30,31,32,35,36,37,38,39,43,44,46,47], of which the main bias presented by the studies was the reporting of results in 7 of 21 studies, and the other items presented a low risk of bias (Table 5 and Figure 6).

### 3.4. Clinical Implications

In TMJ, various symptoms or alterations of the joint have been studied with variants such as the BMC or other variants of morphology at the condylar level that can cause symptoms, such as clicking, ankylosis, or pain associated with the joint. This study investigated the frequency or prevalence of bifurcation of the mandibular condyle with its clinical implications. Of all the studies analyzed, only 10 showed a relationship between the BMC and clinical alterations of the TMJ or surrounding structures. All these studies will be detailed below. In the study by Haghnegahdar et al. [22], a third of the 35 cases of bifid mandibular condyle detected presented clicking, associated pain, or both, but these symptoms only occurred at an advanced age in life without previous symptoms. Knowing that this variant is present for life, this study does not report any traumatic factors triggering the symptoms, which is why the only relationship with the symptoms is advanced age. On the other hand, in the study by Sahma et al. [25], of the ten patients with the BMC, two reported a history of facial trauma due to traffic accidents and clicking when opening their mouth. In the study by Gunduz et al. [25], two patients reported clicking when opening their mouth and a history of trauma, so although the BMC was analyzed as a persistent variant over time, the trigger for the symptoms was trauma.

For the study by Perez et al. [45], the clinical examination of the cases of detected bifid mandibular condyle associated it with limitation in mouth opening, moderate pain, and joint sounds in the temporomandibular joint. In addition to joint sounds, movement restriction, or ankylosis, was also reported, progressing without prior detection of the BMC. In the Rehman et al. [46] study, ten cases of bifid mandibular condyle were detected; all presented with ankylosis, nine reported a history of trauma, and one reported a history of infection causing facial deformity. In the study by Michalski et al. [30], the case of a nine-year-old male child is presented. His medical history does not report any history of trauma or infection, but ankylosis of the temporomandibular joint is detected with difficulty in opening the mouth. Mandibular asymmetries associated with prognathism, retraction, or mandibular deviations also occur. In the study by Lee et al. [30], 28 patients with asymmetric mandibular prognathism and 23 patients with symmetrical mandibular prognathism were identified based on differences in bilateral condyles. This study is also associated with the research by Anzola et al. [46]. In this case, a progressive unilateral condylar growth was found that caused a difference in the global elongation of the condylar neck and the body and ramus of the mandible, presenting facial asymmetry in 16 women and 9 men. Finally, another clinical implication reported in some publications corresponds to malocclusion. In the study by Balaji et al. [31], six patients had limitations in mouth opening accompanied by joint sounds (four males and two females). Finally, in the study by Katti et al. [42], a case of a male patient with limited mandibular opening was reported. What is mentioned in these articles means that in the presence of a BMC, alterations will occur in the normal mobility of the TMJ, especially in the closing movement if the presence of a BMC is unilateral, which in turn produces muscle imbalances, especially of the lateral and medial pterygoid muscles. Finally, the presence of BMC can also be associated with TMJ pain which is accentuated in the presence of TMJ ankylosis.

## 4. Discussion

This systematic review and meta-analysis aimed to report the prevalence of BMC variants and their association with pathologies of the TMJ, infratemporal region, pterygoid muscles, and capsular-ligamentous complex of the temporomandibular region. A prevalence calculation was performed for studies that met the eligibility criteria set by the research team. Using the inclusion criteria, this review attempted to elucidate the characteristics of different anatomical variants of the mandibular condyle. The main finding of our review was that the prevalence of the BMC variant was very low—less than 1%—which correlates with the scientific literature regarding the BMC. Regarding the clinical literature, there is research that shows a relationship, with a greater probability of having some type of chronic and acute symptoms in the TMJ or the infratemporal region.

Other articles have clinically associated the anatomical variants of the BMC with different pathologies of the TMJ. Our review presents a detailed anatomical and clinical approach to the BMC, using updated terminology of the anatomical structures that make up the TMJ. We also provide a functional description and a brief description of the pathophysiology of temporomandibular disorders. It should be noted that we have not found any systematic review and meta-analysis of the BMC, and in the last review with a clinical case, five years have passed since the last anatomical review of the BMC. Regarding the aforementioned reviews, the one by Borras et al. (2018) [48] showed that the BMC can have a congenital or traumatic etiology; hypomobility and arthralgia are the most frequent symptoms, and the treatment options are usually conservative. Unlike this review, we exhaustively detail the anatomy of the mandibular condyle; we also believe that it is necessary to detail whether the condylar variant is a condition of fetal development or one that was caused during life associated with a traumatic event. For the review by Sonneveld et al. (2018) [49], we showed that the BMC is an important anatomical variation that has implications in any mandibular surgery, including implant surgery. A little over 1% of patients have this variation, but not recognizing it in a patient can lead to a bad result. Unlike this study, we show the prevalence and a more detailed anatomy of the variant in addition to saying that the variant can be a structural alteration that, when present, will cause pain and ankylosis, among other symptoms. In the review by Lopez et al. (2010) [50], it is suggested that additional tests, such as MRI or CT, be performed only in cases where the therapeutic approach involves active treatment. It is proposed that the bifid condyle be described as having two condylar heads emerging from the neck of the condyle or below. Unlike our review, these studies only show the anatomy without talking about the prevalence, and they do not give any type of clinical correlation. Finally, the review by Ayat et al. (2019) [51] showed that the anomaly of the mandibular condyle has been described as a condition of unknown etiology and uncertain pathogenesis. Some authors see it as the result of accidental trauma or forceps delivery, with the two heads one behind the other in the sagittal plane. The reported cases are mostly unilateral and generally asymptomatic; unlike what we found, it was mainly not demonstrated that there is no difference between the unilateral and bilateral BMC. In the context of geographical distribution, most of the included studies were conducted in Asia and Europe, but the largest number of samples in the analyzed cohort was from South America. This is a limitation, although we conducted an expanded literature search. Reports were missing from Africa and Oceania, so a more homogeneous geographical distribution was not possible. Consequently, we could not infer whether the BMC is influenced by ethnic factors; however, the included reviews and studies did not show a relationship between ethnic factors that may predispose to the appearance of the BMC. There was no study that showed that variants of the mandibular condyle occurred only unilaterally and bilaterally, which could be more prevalent if it were bilateral in relation to the presence of pathologies such as ankylosis or temporomandibular dysfunctions. Regarding the sex of the subjects who presented the BMC, there was no type of difference between female and male sex, which is why this variant does not present the relationship between presence and sex. Moreover, we have not found any type of study that shows the relationship between sex and BMC. Regarding the prevalence of the BMC, the studies that met the inclusion criteria presented a prevalence of 1%. The literature shows that the prevalence is between 1% and 3%; therefore, what we report in this meta-analysis correlates with previous primary studies. It is possible that we have overestimated or underestimated the prevalence, because the included studies specifically selected patients with the BMC or left out a study that changes these values; however, this was not reported in any of the articles analyzed. Regarding bias, the included case studies and observational studies presented a low risk of bias; however, it should be noted that the latter had a greater bias in the outcome measures. The heterogeneity of the included studies means that the reported data must be taken with caution, and it is also proposed to carry out more studies that establish the association between anatomical variants and clinical implications. This was not detailed in any study though, which is important because there could be overestimated or underestimated data in their conclusions. Regarding the clinical considerations related to the presence of the BMC, the most associated and evidently studied biomechanical phenomenon is the alteration in the fit of the accessory region of the condyle, which causes the joint to be altered during opening and closing movements. This produces a closing movement that is mainly altered because the presence of the BMC often causes there to be an anticipated stop between the articular surfaces of the condyle and the mandibular fossa. If the presence is bilateral, the mechanizing literature reports that the movements in the sagittal plane will be altered and could be more symptomatic in these patients. If the BMC is unilateral, the functional and mechanical alteration will be mainly the lateral movement, and it is reported that the dysfunction and symptoms will mainly depend on the length of the BMC; if it is longer, symptomatology will be ipsilateral, while if it is shorter, the symptomatology will be contralateral. Finally, in this condition of unilaterality, the functional alteration will be contralateral to the presence of the BMC. In the presence of these symptoms and ruling out other possible pathologies that are more prevalent, it is important to perform imaging studies that allow the observation of the BMC. The importance of early diagnosis lies in early detection, and more importantly, in preventing the development of functional and symptomatic alterations in the TMJ. The main symptoms that could suggest the presence of the BMC are pain in the joint space of the TMJ, clicking when moving(which can be painless or with the presence of symptoms in the TMJ),and finally, a symptom that is also repeatedly described in the articles, namely, ankylosis of the TMJ, which presents with little movement and problems in the chewing process and speech of the subjects studied. Finally, it should be noted that, although these are cardinal signs in the face of other pathologies, the presence of these could prompt professionals to analyze the joint through imaging and be able to see the presence of the BMC [52,53]. The symptoms are also varied, but a significant group of studies showed that this anatomical variant could also present asymptomatically throughout life, going unnoticed by many patients.

## 5. Limitations

The limitations of this review were the publication and authorship bias of the included studies. Studies with different results that were in non-indexed literature in the selected databases may have been excluded, and there is the possibility that the most sensitive and specific search regarding the topic to be studied was not carried out. Finally, the individual sessions of the authors for the selection of articles all resulted in a higher probability of excluding potential cases that are not being reported in the scientific community from countries other than those on the Asian and European continents.

## 6. Conclusions

In the present study, we have found a correlation between BMC and TMJ pathologies. Through a meta-analysis, we found that the presence of BMC has a relatively low prevalence in studies that presented a considerable number of subjects. From a clinical point of view, a direct association between the presence of BMC and TMJ pathologies or symptoms cannot be established. Considering the above, we believe that knowing this variant is of utmost importance for dental surgeons, especially for those who treat the TMJ region, since it is important to have in-depth knowledge to generate the best guidelines for the treatment and diagnosis of this type of pathology. We also recommend that in the presence of BMC, the masticatory muscles be dynamically worked unilaterally or bilaterally depending on how the BMC presents, in addition to therapies for the symptoms of TMJ. Finally, we believe that it is important to carry out new anatomical and clinical studies that clearly define this condition in the jaw.

## Figures and Tables

**Figure 1 diagnostics-13-03282-f001:**
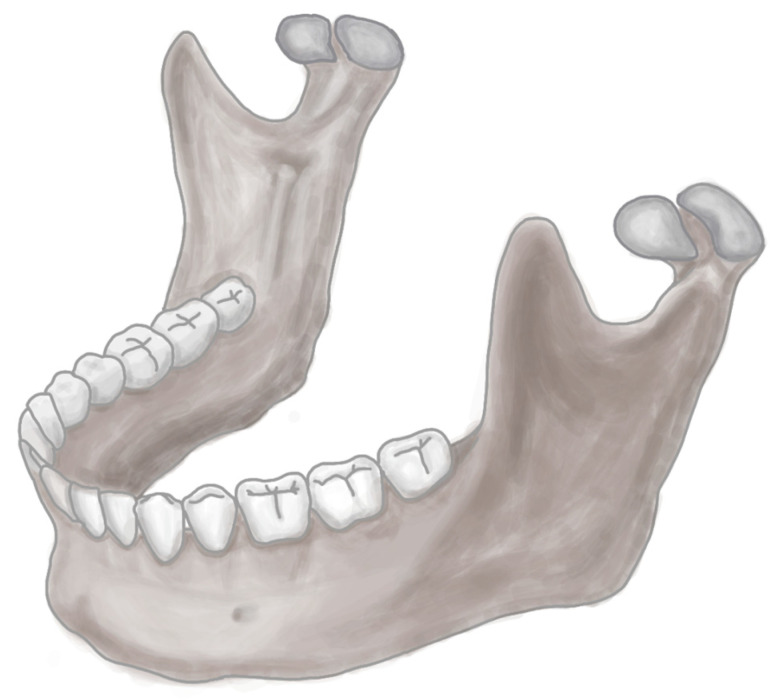
BMC.

**Figure 2 diagnostics-13-03282-f002:**
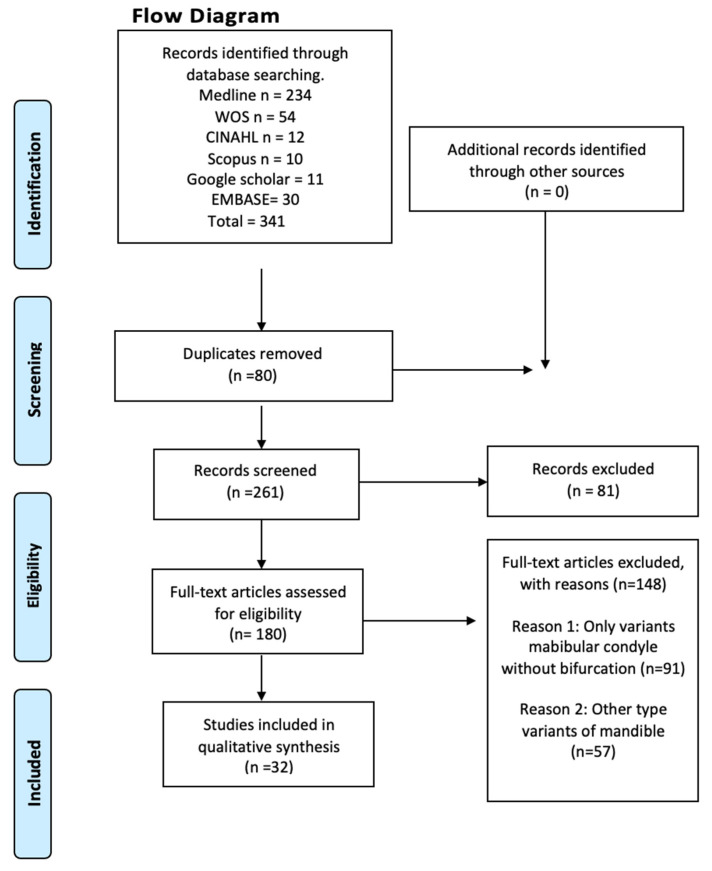
Search diagram.

**Figure 3 diagnostics-13-03282-f003:**
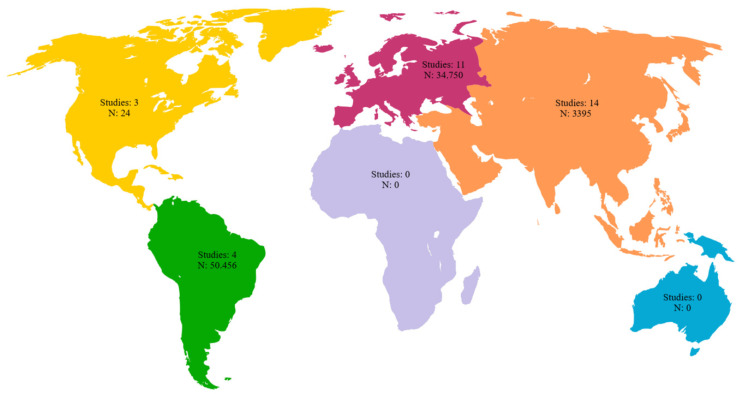
Geographic distribution of the included studies with number of subjects per continent.

**Figure 4 diagnostics-13-03282-f004:**
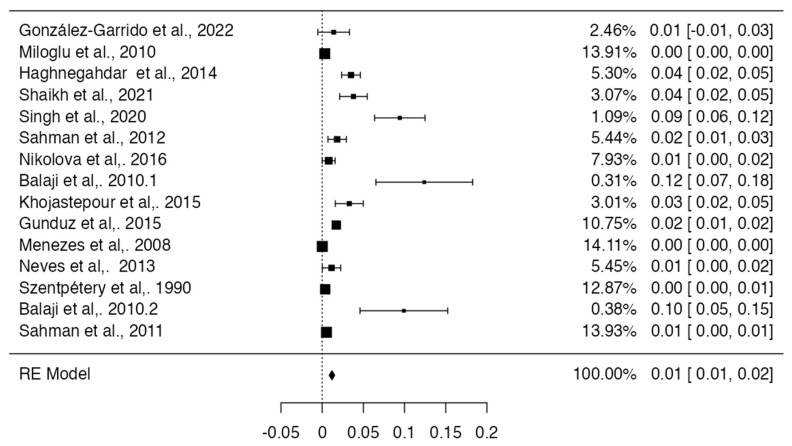
Forest plot prevalence of the included studies with BMC [16,17,22,23,24,25,27,31,32,35,36,37,38,39,47].

**Figure 5 diagnostics-13-03282-f005:**
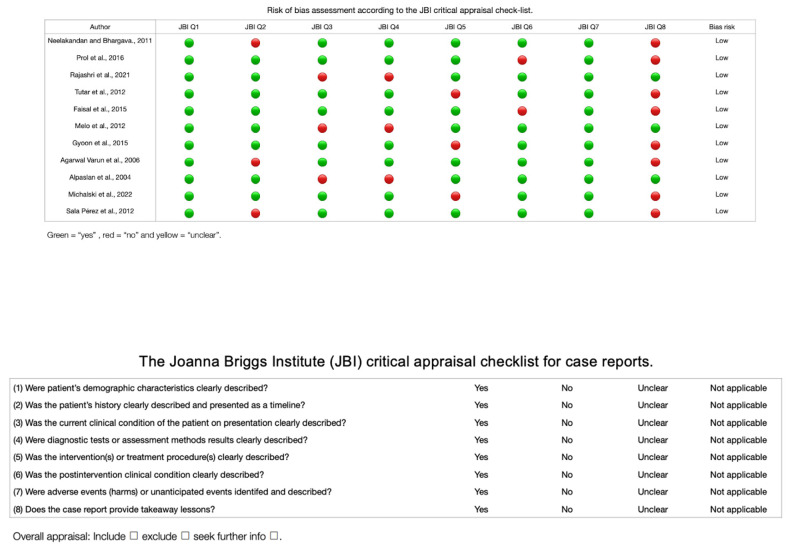
Risk of bias graph of the included case studies [18,19,20,21,28,29,33,34,40,41,45].

**Figure 6 diagnostics-13-03282-f006:**
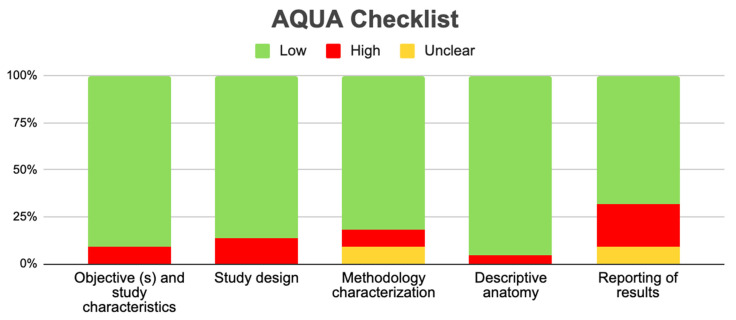
Risk of bias graph of the included observational studies.

**Table 1 diagnostics-13-03282-t001:** Characteristics of the included studies.

Author and Year	Type Study and *N*	Prevalence and Characteristics of Subjects	Static Values	Region	Sex of Samples	Laterality	Clinical Considerations
González-Garrido et al., 2022 [16]	Adult mandibles and cranial bone (143)	Two (1.4%) mandibles with both condyles exhibit multi-headed condyle	No present static values	Spain	141 sex unknown; 1 female; 1 male	Unilateral	Both cases of the present study are post-traumatic, no deterioration was identified, severe mandibular use or malfunction, with temporomandibular joint osteoarthritis and eversion of the gonial angle.
Miloglu et al., 2010 [17]	Radiographs from patients undergoing dental treatment (10,200)	32 (0.3%) had bifid mandibular condyle	No present static values	Turkey	10,168 unknown gender; 17 female; 15 male.	32 patients, 24 (75.0%) had unilateral, 8 (25.0%) had bilateral	This anomaly does not present any clinical symptoms and dentists are more interested in dental pathologies in the examination of radiographs.
Neelakandan and Bhargava., 2011 [18]	Panoramic radiography and computerized tomography evaluation of chin deviation on a 14-year-old male (1)	Unique in study	No present static values	India	One male	Unilateral	First reported case of bifid mandibular condyle with condyle hyperplasia. The etiology of bifid condyle is largely unknown, although various factors have been suggested as possible causes like endocrine.
Prol et al., 2016 [19]	13-year-old female orthopantomography and palpation in masticatory muscles (1)	Unique in study	No present static values	Spain	One female	Bilateral	The authors do not refer to previous trauma. Both upper condylar surfaces with a depression, compatible with bifid condyle of mediolateral type.
Rajashri et al., 2021 [20]	Magnetic resonance evaluation of a 38-year-old male patient (1)	Male patient with bifid mandibular condyle	No present static values	India	One male	Unilateral	His magnetic resonance showed signs of mild degenerative changes of the bilateral articular disk with reduced translation on the right side and a bifid left mandibular condyle with a small cyst next to the left temporomandibular joint.
Tutar et al., 2012 [21]	Panoramic radiographs of a 24-year-old patient (1)	One patient with bifid mandibular condyle	No present static values	Turkey	One female	Bilateral	Most cases are asymptomatic, but there can be associated symptoms such as clicking, moderate pain, and limitation of mandibular movements.
Haghnegahdar AA et al., 2014 [22]	Dental panoramic views of individuals (1000)	Bifidity was detected in 35 cases (3.5%).	No present static values	Iran	767 female and 233 male individuals. 23 female patients and 12 male patients showed bifidity.	35 patients; 32 had unilateral (24 on the left and 8 on the right side) and 3 had bilateral bifid mandibular condyle.	Approximately one third of the cases of the study were symptomatic, suffering from clicking, pain, or both. This anomaly may be misinterpreted as the presence of tumors or fractures in the condylar area.
A.H.Shaikh et al., 2021 [23]	Panoramic radiographic evaluation of 500 mandibular condyles (250)	Different shapes of condyles have been identified, namely, oval (50%), bird beak (40%), diamond (4.8%) and crooked finger (4.8%) shape.	No present static values	Pakistan	125 male patients and 125 female patients.	Not mentioned	Other studies reported that the condyle morphology with changes in condyle surface shapes is related to malocclusion and the relation between open bite and erosion of the head of the condyle.
Bhupender Singh et al., 2020 [24]	Examination of panoramic radiographs of mandibular condyles (350)	Dentition status was classified using the Eichner index. Eichner class A: 282 persons; Eichner class B: 33 persons; Eichner class C: 35 persons.	The relation between age groups and denture usage history was statistically significant (*p* = 0.00).	India	155 male patients (44%) and 195 female patients (56%).	Not mentioned	There is a relation between dentition status and bilaterally similar condylar morphology.
Halil Sahman et al., 2012 [25]	Retrospective study of computerized tomography records (550)	This anomaly was found in 10 patients of the 550 (1.82%). A total of 13 bifid mandibular condyles were found in these 10 patients.	No significant gender difference in subjects with Bifid mandibular condyle (*p* > 0.5) No significant difference between right or left side bifid mandibular condyle (*p* > 0.5)	Turkey	328 (59.6%) male patients and 222 (40.4%) female patients. 5 female patients and 5 male patients showed bifidity.	Three patients had bilateral and seven patients had unilateral bifid mandibular condyle (three on the left side and four on the right side).	Six patients were contacted. Two of them had a history of head trauma, reporting clicking on mouth opening and bilateral temporomandibular joint pain.
Rehman et al., 2009 [26]	Examination of computerized tomography of patients with temporomandibular joint ankylosis (37)	Of the 37 patients with temporomandibular joint ankylosis, 10 had bifid mandibular condyle.	No present static values	India	Of the 10 patients, 5 were male and 5 were female.	- Two patients had bilateral mediolaterally bifid mandibular condyle with unilateral temporomandibular joint ankylosis. - Two patients had bilateral anteroposteriorly bifid mandibular condyle. One of them had bilateral temporomandibular joint ankylosis and the other unilateral temporomandibular joint ankylosis. - Six patients had unilateral mediolaterally bifid mandibular condyle with ipsilateral temporomandibular joint ankylosis.	Nine patients report a history of trauma and one patient reports a history of infection. Among the nine patients that report a history of trauma, eight sustained falls on the face and one sustained a road traffic accident with penetrating facial injury. One patient refers that when he was 6 years old he had an infection episode, developing facial deformity and restriction of mouth opening.
Nikolova et al., 2016 [27]	Macroscopic observation of the condyles of dry intact mandibles from adult males (500)	Four patients present bifid mandibular condyle (0.8%).	No present static values	Bulgaria	All 500 patients were males; 4 showed bifidity.	All of the bifid mandibular condyle cases were unilateral, two on the right side and two on the left side.	Case 1 presents osteoarthritis in the mandibular fossa. Case 2 presents a shallow depression on the left condyle.
Faisal et al., 2015 [28]	Computerized tomography examination of two cases (2)	Two patients present bifid mandibular condyle.	No present static values	India	Both patients were female.	Both cases were unilateral bifid mandibular condyles. Case 1 presents left bifid mandibular condyle oriented anteroposteriorly. Case 2 presents right bifid mandibular condyle oriented mediolaterally.	Bifid mandibular condyle appears to be more common on the left side in unilateral cases (2:1).
Melo et al., 2012 [29]	Magnetic resonance of a 39-year-old female patient with mouth opening limitation and deviation of the mandible to the left side (1)	One female patient with bifid mandibular condyle and duplicated mandibular fossa, with the articular disc over the anterior head	No present static values	Brazil	Female patient	Unilateral bifid mandibular condyle present on the left side and duplicated mandibular fossa	A diagnosed mouth opening limitation and deflection of the mandible to the left side. This is the only case of bifid mandibular condyle that includes an anteroposterior bifid condyle.
Lee JS et al., 2017 [30]	Evaluation of bilateral difference in condyle position of patients with deviated mandibular prognathism using 3D reformatted images from cone beam computerized tomography (51)	28 patients with asymmetric mandibular prognathism; 23 patients with symmetric mandibular prognathism.	Differences in the position of lateral condyle. More laterally and inferiorly in the contralateral side. (*p* < 0.05) differences in the position of the sigmoid notch. More laterally, superiorly and posteriorly positioned on the deviated side (*p* < 0.01)	Republic of Korea	16 female patients with asymmetric mandibular prognathism and 9 female patients with symmetry; 12 male patients with asymmetric mandibular prognathism and 14 with symmetry.	Bilateral condylar position study	Asymmetric mandibular prognathism
Balaji et al., 2010 [31]	Computed tomograms performed on patients with temporomandibular joint ankylosis (121)	Of all 121 cases, 15 were diagnosed with bifid mandibular condyle (12.40%).	Difference between deviation of chin of bilateral bifid mandibular condyle (*p* = 0.000)	India	Seven male patients with bifid mandibular condyle; eight female patients with bifid mandibular condyle.	Four cases had bilateral bifid mandibular condyle. Eleven cases had unilateral bifid mandibular condyle.	All of the 15 cases reported in the study were oriented mediolaterally and all cases with mandibular joint ankylosis.
Khojastepour et al., 2015 [32]	Evaluation of patients’ cone beam computerized tomography scans to evaluate prevalence of bifid mandibular condyle (425)	309 of 425 patients entered in the study due to acceptable visibility of condyles. 14 cases of bifid mandibular condyle were detected (4.53%).	No present static values	Iran	Of the 309 patients in the study, 170 were female (55%) and 139 were male (45%); 7 female patients and 7 male patients showed bifidity.	3 had bilateral bifid mandibular condyle; 11 had unilateral bifid mandibular condyle, 5 cases were detected on the right side and 6 were detected on the left side.	The use of cone beam computerized tomography scans to evaluate temporal mandibular joint area has the advantage that it eliminates superimpositions in the images.
Gyoon et al., 2015 [33]	Use of nanoindentation in human cadavers to examine variations of the elastic, plastic, and viscoelastic mechanical properties of human mandibular condyle bone tissue. (9)	Cortical and trabecular bone dissected from mandibular condyles of nine human cadavers	Significant difference between high gray values of endosteal cortical bone and periosteal cortical bone and trabecular bone (*p* < 0.007)	USA	9 male patients	Not mentioned	This is the first study that measures five parameters of elastic, plastic, and viscoelastic mechanical properties in fresh human mandibular condylar bone using nanoindentation.
Kaan Gunduz et al., 2015 [34]	Cone beam computed tomography images of patients to study the frequency of bifid mandibular condyle (2634)	Of the 2634 patients of the study, 45 bifid mandibular condyles were found in 42 (1.7%) patients.	No significant difference of clinical symptoms between patients with normally shaped condyles and bifid mandibular condyle (*p* > 0.05)	Turkey	Of the 2634 patients, 1455 (45.41%) were male and 1179 (49.54%) were female, and of the 42 patients with bifid mandibular condyle, 22 (52.38%) were male and 20 (47.62%) were female.	Of all 45 bifid mandibular condyles, 39 (92.8%) were unilateral cases and 3 (7.1%) bilateral cases. 24 cases (53.3%) were on the right side and 21 cases (46.6%) were on the left side.	Two patients reported a history of trauma and clicking on mouth opening.
Menezes et al., 2008 [35]	Examination of radiographic images in a group of patients to evaluate the morphology and frequency of bifid mandibular condyles (50,080)	Of all 50,080 panoramic radiographs, only 9 (0.018%) cases of bifid mandibular condyle were found.	No present static values	Brazil	Seven female and two male patients with bifid mandibular condyle.	Seven unilateral cases. Four cases on the left side and three on the right side. Two bilateral cases.	There were no cases of history of previous trauma, pain or trismus.
Neves et al., 2013 [36]	Comparison between panoramic radiography and cone beam computerized tomography of individuals (350)	Of all 350 individuals, 4 (1.1%) cases of bifid mandibular condyle were detected.	No present static values	Brazil	Three female patients and one male patient with bifid mandibular condyle.	All cases were unilateral. Three of them were detected on the right side and one was detected on the left side.	In all four cases, a history of trauma was reported and the relation of one condylar process to the other was mediolateral.
Szentpétery et al., 1990 [37]	Examination of prehistoric skulls with a total number of condyles (1882)	Among the 1882 skulls, 7 cases of bifid mandibular condyle were detected.	No present static values	Hungary	Five female individuals and two male individuals with bifid mandibular condyle.	The seven cases of bifid mandibular condyle were unilateral cases. Two were detected on the left side and five on the right side.	In the seven cases, the grooving was anteroposteriorly directed.
Balaji et al., 2010 [38]	Retrospective examination of patients computerized tomography (121)	Of all 121 cases, 12 patients presented bifid mandibular condyle.	No present static values	India	Five male and seven female patients were detected with bifid mandibular condyle.	Three bilateral cases and nine unilateral cases. Eight were detected on the left side and one on the right side.	All cases of the study exhibited mushroom-shaped bifid condyle. History of trauma was reported in 91.7% of cases.
Halil Sahman et al., 2011 [39]	Retrospective study of panoramic radiographs (18,798)	Of all 18,798 cases, 98 patients were detected with bifid mandibular condyle.	No statistically significant differences between right and left bifid mandibular condyles, or between female and male patients (*p* > 0.05).	Turkey	51 female patients and 47 male patients with bifid mandibular condyle.	27 bilateral cases and 71 unilateral cases with bifid mandibular condyle. 37 cases were detected on the right side and 34 cases on the left side.	The frequency of bifid mandibular condyle is higher.
Agarwal Varun et al., 2006 [40]	Panoramic radiograph of a bifid mandibular condyle. They reported four cases: two patients and two in archived specimens. (48)	The condylar head is duplicated (dividing it into medial and lateral condylar heads) (4/48).	No present static values	India	Two specimens Two female	Two bilateral bifid condyles Two right-sided bifid mandibular condyle	In the first case, a female patient referred to a polyarthralgia before she reported her problem with a limited mouth opening. She referred only to malocclusion. In the other case, the patient has a bifid mandibular condyle, with no pain and normal mouth opening.
Alpaslan et al., 2004 [41]	A 40-year-old male with pain at the both temporomandibular joints. In a routine dental examination with a panoramic radiograph revealed bilateral bifid condyles. (1)	Unique in study	No present static values	Turkey	One male	Bilateral	The patient referred to a moderate pain during chewing at the bilateral temporomandibular joint. His maximum opening was 48 mm.
Katti et al., 2012 [42]	Panoramic radiography of a 20-year-old male patient with limited mouth opening and cosmetic disfigurement (1)	Unique case in study	No present static values	India.	One male	Right side	The patient presented a limitation of jaw movement and his mouth opening was limited to 27 mm. The examination of the head revealed a mandibular micrognathia.
Michalski et al., 2022 [34]	Nine-year-old patient with unilateral ankylosis of the temporomandibular joint (TMJ). In a physical examination a deviation of the mandible was noted. To evaluate they used computed tomography imaging (9)	Unique case in the study	No present static values	USA	One male	Left side	The patient did not refer to a trauma or infection in his clinical history, but the ankylosis of the temporomandibular joint (TMJ) was a defect to development of bifid mandibular condyle, with difficulty opening his mouth.
Screenivasagan et al., 2021 [43]	Evaluation of radiographs of condylar heads by an orthopantomogram (987)	1048 oval, 148 crooked, 382 bird beak, and 396 diamond condylar morphology	No present static values	India	512 female 475 male	Not mentioned	The anatomical morphology defines the progression of symptoms and the occlusion of the mandible.
Schmitter et al., 2006 [44]	Magnetic resonance imaging evaluation (40)	21 Patients complained about arthrogenic problems and 19 patients did not complain about arthrogenic problems.	No present static values	Germany	13 male 27 female	Not mentioned	Temporomandibular disorder often involves the action of masticatory muscles. There were symptomatic and asymptomatic patients.
Sala Pérez et al., 2012 [45]	Panoramic radiography was the technique of diagnosing bifid mandibular condyle. In this study, cases with anatomical variation of the temporomandibular joint were analyzed. (6)	Microtrauma or trauma, malocclusion, radiotherapy or infections; all of these factors may produce alterations in the condylar joint.	No present static values	USA	Four male Two female	One case with bilateral bifid mandibular condyle. Five cases with unilateral bifid mandibular condyle (right)	Clinical examination revealed a limitation in mouth opening, moderate pain, and joint sounds in the temporomandibular joint (TMJ).
Anzola et al., 2021 [46]	This study investigated normal activity values of the mandibular condyles by bone scintigraphy. (25)	Characterized by progressive unilateral growth, resulting in global enlargement of the condyle including the condylar neck and the body and ramus of the jaw	No present static values	Colombia	16 female 9 male	Not mentioned	Facial asymmetry and occlusal alterations
Hiperplasia condilar							

**Table 2 diagnostics-13-03282-t002:** Methods for identifying the BMC in the subjects included in this study.

Diagnostic Method	Number of Articles	Total Number of Subjects
Macroscopic evaluation (corpse)	3	2525
Magnetic resonance	3	42
CT scan	6	840
Cone beam computerized tomography	3	3110
Orthopantomography	2	988
Panoramic X-ray	13	81,086
Bone scintigraphy	1	25
Nanoindentation	1	9

**Table 3 diagnostics-13-03282-t003:** Prevalence of included observational articles.

Author	Total *N*	Prevalence	Prevalence Meta-Analysis Status
González-Garrido et al., 2022 [16]	143	2	Included
Miloglu et al., 2010 [17]	10,200	32	Included
Haghnegahdar et al., 2014 [22]	1000	35	Included
Shaikh et al., 2021 [23]	500	19	Included
Singh et al., 2020 [24]	350	33	Included
Sahman et al., 2012 [25]	550	10	Included
Nikolova et al., 2016 [27]	500	4	Included
Lee et al., 2017 [30]	51	23	Not included
Rehman et al., 2009 [26]	37	10	Not included
Balaji et al., 2010 [31]	121	15	Included
Khojastepour et al., 2015 [32]	425	14	Included
Gunduz et al., 2015 [47]	2634	45	Included
Menezes et al., 2008 [35]	50,080	9	Included
Neves et al., 2013 [36]	350	4	Included
Szentpétery et al., 1990 [37]	1882	7	Included
Balaji et al., 2010 [38]	121	12	Included
Sahman et al., 2011 [39]	18,798	98	Included
Sreenivasagan S et al., 2021 [43]	48	4	Not included
Schmitter et al., 2006 [44]	40	19	Not included
Anzola et al., 2021 [46]	25	2	Not included

**Table 4 diagnostics-13-03282-t004:** Risk of case study bias.

Author	JBI Q1	JBI Q2	JBI Q3	JBI Q4	JBI Q5	JBI Q6	JBI Q7	JBI Q8	Bias Risk
Neelakandan and Bhargava, 2011 [18]	Yes	No	Yes	Yes	Yes	Yes	Yes	No	Low
Prol et al., 2016 [19]	Yes	Yes	Yes	Yes	Yes	No	Yes	No	Low
Rajashri et al., 2021 [20]	Yes	Yes	No	No	Yes	Yes	Yes	Yes	Low
Tutar et al., 2012 [21]	Yes	Yes	Yes	Yes	No	Yes	Yes	No	Low
Faisal et al., 2015 [28]	Yes	Yes	Yes	Yes	Yes	No	Yes	No	Low
Melo et al., 2012 [29]	Yes	Yes	No	No	Yes	Yes	Yes	Yes	Low
Gyoon et al., 2015 [33]	Yes	Yes	Yes	Yes	No	Yes	Yes	No	Low
Varun et al., 2006 [40]	Yes	No	Yes	Yes	Yes	Yes	Yes	No	Low
Alpaslan et al., 2004 [41]	Yes	Yes	No	No	Yes	Yes	Yes	Yes	Low
Michalski et al., 2022 [34]	Yes	Yes	Yes	Yes	No	Yes	Yes	No	Low
Pérez et al., 2012 [45]	Yes	No	Yes	Yes	Yes	Yes	Yes	No	Low

**Table 5 diagnostics-13-03282-t005:** Risk of bias in observational studies (AQUA).

References	Study Design	Domain 1				Domain 2					Domain 3						Domain 4					Domain 5				
		1	2	3	4	5	6	7	8	9	10	11	12	13	14	15	16	17	18	19	20	21	22	23	24	25
González-Garrido et al., 2022 [16]	Observational study	Y	Y	Y	Y	Y	Y	Y	Y	Y	Y	N	Y	N	Y	Y	N	Y	Y	NA	Y	NA	Y	Y	N	Y
Miloglu et al., 2010 [17]	Observational study	N	Y	Y	Y	Y	Y	Y	Y	Y	Y	N	N	N	Y	N	Y	Y	Y	Y	Y	Y	Y	Y	Y	Y
Haghnegahdar AA et al., 2014 [22]	Observational study	Y	Y	N	N	Y	Y	Y	Y	Y	Y	N	N	N	Y	Y	Y	Y	Y	Y	Y	NA	Y	Y	Y	Y
A.H.Shaikh et al., 2021 [23]	Observational study	Y	Y	Y	Y	Y	Y	Y	Y	Y	N	N	Y	N	Y	Y	Y	Y	Y	Y	Y	NA	Y	Y	Y	Y
Bhupender Singh et al., 2020 [24]	Observational study	Y	N	Y	N	Y	Y	Y	Y	Y	Y	N	Y	Y	Y	Y	Y	Y	Y	Y	Y	NA	Y	Y	Y	Y
Halil Sahman et al., 2012 [25]	Observational study	Y	Y	N	Y	Y	Y	Y	Y	Y	Y	N	N	N	Y	Y	Y	Y	Y	NA	Y	NA	Y	Y	Y	Y
Nikolova et al., 2016 [27]	Observational study	Y	Y	Y	Y	Y	Y	Y	Y	Y	N	N	N	N	Y	Y	Y	Y	Y	NA	Y	NA	Y	Y	N	Y
Lee JS et al. 2017 [30]	Observational study	Y	Y	Y	Y	Y	Y	Y	Y	Y	N	N	N	N	Y	Y	N	Y	Y	NA	Y	NA	Y	Y	N	Y
Rehman et al., 2009 [26]	Observational study	N	Y	Y	Y	Y	Y	Y	Y	Y	Y	N	N	N	Y	N	Y	Y	Y	Y	Y	Y	Y	Y	Y	Y
Balaji et al., 2010 [31]	Observational study	N	Y	Y	Y	Y	Y	Y	Y	Y	Y	N	N	N	Y	N	Y	Y	Y	Y	Y	Y	Y	Y	Y	Y
Khojastepour et al., 2015 [32]	Observational study	Y	Y	Y	Y	Y	Y	Y	Y	Y	Y	N	N	Y	Y	Y	N	N	Y	N	N	NA	Y	Y	Y	Y
Kaan Gunduz et al., 2015 [34]	Observational study	Y	Y	Y	Y	Y	Y	Y	Y	Y	Y	N	N	N	Y	Y	Y	Y	Y	NA	Y	NA	Y	Y	Y	Y
Menezes et al., 2008 [35]	Observational study	Y	Y	Y	Y	N	Y	Y	Y	N	Y	N	Y	Y	N	Y	Y	Y	Y	N	Y	Y	Y	N	Y	NA
Neves et al., 2013 [36]	Observational study	N	Y	N	Y	Y	Y	Y	Y	Y	N	N	N	N	Y	Y	N	Y	Y	NA	Y	NA	Y	Y	N	Y
Szentpétery et al., 1990 [37]	Observational study	Y	Y	Y	Y	Y	Y	Y	Y	Y	Y	N	N	Y	Y	Y	N	N	Y	N	N	NA	Y	Y	Y	Y
Balaji et al., 2010 [38]	Observational study	Y	Y	Y	Y	Y	Y	Y	Y	Y	Y	N	N	N	Y	Y	Y	Y	Y	NA	Y	NA	Y	Y	Y	Y
Halil Sahman et al., 2011 [39]	Observational study	Y	Y	Y	Y	Y	Y	Y	Y	Y	Y	N	Y	N	Y	Y	N	Y	Y	NA	Y	NA	Y	Y	N	Y
Sreenivasagan S et al., 2021 [43]	Observational study	N	Y	Y	Y	Y	Y	Y	Y	Y	Y	N	N	N	Y	N	Y	Y	Y	Y	Y	Y	Y	Y	Y	Y
Schmitter et al., 2006 [44]	Observational study	Y	Y	Y	Y	Y	Y	Y	Y	Y	N	N	N	N	Y	Y	Y	Y	Y	NA	Y	NA	Y	Y	N	Y
Rehman et al., 2009 [26]	Observational study	N	Y	Y	Y	Y	Y	Y	Y	Y	Y	N	N	N	Y	N	Y	Y	Y	Y	Y	Y	Y	Y	Y	Y
Anzola et al., 2021 [46]	Observational study	Y	Y	Y	Y	Y	Y	Y	Y	Y	Y	N	Y	N	Y	Y	N	Y	Y	NA	Y	NA	Y	Y	N	Y

## Data Availability

Not applicable.

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
