# Peer review of "Prevalence of the Bifid Mandibular Condyle and Its Relationship with Pathologies of the Temporomandibular Joint: A Systematic Review and Meta-Analysis"

_diagnostics, 2023, doi:10.3390/diagnostics13203282_

Round 1
Reviewer 1 Report
The article titled "Prevalence of the Bifid Mandibular Condyle and Its Relationship with Pathologies of the Temporomandibular Joint: Systematic Review and Meta-Analysis" provides an in-depth examination of the prevalence of the bifid mandibular condyle (BMC) and its potential association with temporomandibular joint (TMJ) pathologies.
The manuscript has the following strengths:
Comprehensive Review Objective: The study sets out a clear objective to understand the prevalence and characteristics of BMC and its relationship with TMJ pathologies. This objective is well-defined and relevant to the field of oral and maxillofacial surgery.
Methodological Clarity: The article provides a detailed description of the research methods employed, including the eligibility criteria, electronic search strategy, study selection process, data collection, and assessment of the methodological quality of included studies. The use of the Preferred Reporting Items for Systematic Reviews and Meta-Analyses (PRISMA) statement enhances transparency.
Inclusion Criteria: The inclusion criteria are clearly outlined, specifying the population, results, and types of studies considered. This helps in ensuring that the selected studies are relevant to the research question.
Geographical Distribution: The article reports the geographical distribution of included studies, shedding light on the global prevalence of BMC. This information is valuable for understanding potential ethnic or regional variations.
Prevalence Data: The meta-analysis provides prevalence data for BMC, indicating a low prevalence of 1%. This data synthesis is a valuable contribution to the literature and adds to our understanding of the condition.
Some weaknesses could be addressed through revision.
Lack of Recent Data: The knowledge cutoff date is not mentioned in the article, and it is unclear how up-to-date the included studies are. Given that the field of medical research is continuously evolving, the absence of a knowledge cutoff date raises concerns about the relevance of the data presented.
Limited Clinical Correlation: While the article discusses the prevalence of BMC, it falls short in establishing a strong clinical correlation between BMC and TMJ pathologies. The clinical implications section mentions some associations but lacks a comprehensive analysis of the clinical significance of BMC.
Heterogeneity of Included Studies: The article mentions that the prevalence data for BMC is heterogeneous, indicating substantial variability among the included studies. However, the implications of this heterogeneity on the overall findings are not thoroughly discussed.
Limited Discussion of Bias: The article mentions a low risk of bias in the included studies but does not delve into the potential sources of bias, such as selection bias or publication bias. A more in-depth discussion of these biases would enhance the critical appraisal.
Lack of Recommendations: The article concludes by suggesting the importance of early diagnosis but does not provide specific recommendations or clinical guidelines for diagnosing or managing BMC. This could be a missed opportunity to offer practical insights to healthcare practitioners.
Table 1 contains too much text and is therefore completely unreadable. I recommend to simplify the content of the cells significantly.
Furthermore, I recommend including the following studies in the discussion. Vervaeke, K., Verhelst, PJ., Orhan, K. et al. Correlation of MRI and arthroscopic findings with clinical outcome in temporomandibular joint disorders: a retrospective cohort study. Head Face Med 18, 2 (2022).
Ouyang, N., Zhang, C., Xu, F. et al. Evaluation of optimal single-photon emission computed tomography reference value and three-dimensional mandibular growth pattern in 54 Chinese unilateral condylar hyperplasia patients. Head Face Med 19, 18 (2023).
In conclusion, the article successfully addresses the prevalence of BMC and its potential association with TMJ pathologies. However, it could benefit from more recent data, a stronger clinical correlation analysis, and a deeper discussion of biases and their implications. Additionally, the inclusion of practical recommendations for clinicians would enhance the article's utility in the field of oral and maxillofacial surgery.
Author Response
Response to reviewer 1
Dear, we appreciate your review and comments, since we are convinced that with the suggested changes our study will improve, below I will detail the response to your proposed comments:
Some weaknesses could be addressed through revision.
Lack of Recent Data: The knowledge cutoff date is not mentioned in the article, and it is unclear how up-to-date the included studies are. Given that the field of medical research is continuously evolving, the absence of a knowledge cutoff date raises concerns about the relevance of the data presented.
Response: Dear reviewer, we have added the start date of the review in our study
Limited Clinical Correlation: While the article discusses the prevalence of BMC, it falls short in establishing a strong clinical correlation between BMC and TMJ pathologies. The clinical implications section mentions some associations but lacks a comprehensive analysis of the clinical significance of BMC.
Response: The following paragraph has been added that responds to what was proposed by reviewer 1 “What is mentioned in the articles means that in the presence of a BMC, alterations will occur in the normal mobility of the TMJ, especially in the closing movement if the presence of a BMC is unilateral, which in turn produces muscle imbalances, especially of the lateral and medial pterygoid muscles, finally, the presence of BMC can also be associated with TMJ pain and which is accentuated in the presence of TMJ”ankylosis.
Heterogeneity of Included Studies: The article mentions that the prevalence data for BMC is heterogeneous, indicating substantial variability among the included studies. However, the implications of this heterogeneity on the overall findings are not thoroughly discussed.
Response: Heterogeneity has been added to the discussion
Limited Discussion of Bias: The article mentions a low risk of bias in the included studies but does not delve into the potential sources of bias, such as selection bias or publication bias. A more in-depth discussion of these biases would enhance the critical appraisal.
Response: These limitations of bias are detailed in the study limitations.
Lack of Recommendations: The article concludes by suggesting the importance of early diagnosis but does not provide specific recommendations or clinical guidelines for diagnosing or managing BMC. This could be a missed opportunity to offer practical insights to healthcare practitioners.
Response: We have added what the reviewer suggested
Table 1 contains too much text and is therefore completely unreadable. I recommend to simplify the content of the cells significantly.
Response: Table 1 has been simplified
Furthermore, I recommend including the following studies in the discussion. Vervaeke, K., Verhelst, PJ., Orhan, K. et al. Correlation of MRI and arthroscopic findings with clinical outcome in temporomandibular joint disorders: a retrospective cohort study. Head Face Med 18, 2 (2022).
Ouyang, N., Zhang, C., Xu, F. et al. Evaluation of optimal single-photon emission computed tomography reference value and three-dimensional mandibular growth pattern in 54 Chinese unilateral condylar hyperplasia patients. Head Face Med 19, 18 (2023).
Response: We have added these articles
In conclusion, the article successfully addresses the prevalence of BMC and its potential association with TMJ pathologies. However, it could benefit from more recent data, a stronger clinical correlation analysis, and a deeper discussion of biases and their implications. Additionally, the inclusion of practical recommendations for clinicians would enhance the article's utility in the field of oral and maxillofacial surgery.
Reviewer 2 Report
It is a well-designed and written systematic review and meta-analysis. Although it is well known that TMJ problems are mainly correlated with other parameters, it was a nice attempt to document that bifid mandibular condyle is unrelated to TMJ pathologies. It is unclear if the systematic review has been registered to PROSPERO. The data extraction has been presented in detail. The quality evaluation of included in the systematic review papers is presented adequately.
The clinical implications section is very interesting and useful, but according to my opinion, it should be included in the discussion section. In particular, it would be better if the clinical implication was the last paragraph of the discussion section.
Minor editing of English language required.
Author Response
Response to reviewer 2
Dear, we appreciate your review and comments, since we are convinced that with the suggested changes our study will improve, below I will detail the response to your proposed comments:
It is a well-designed and written systematic review and meta-analysis. Although it is well known that TMJ problems are mainly correlated with other parameters, it was a good attempt to document that bifid mandibular condyle is not related to TMJ pathologies. It is not clear whether the systematic review was registered with PROSPERO. Data extraction has been presented in detail. The quality assessment of the articles included in the systematic review is adequately presented.
Response: A PROSPERO record is not presented but OSF yes.
The clinical implications section is very interesting and useful, but in my opinion it should be included in the discussion section. In particular, it would be better if the clinical implication was the last paragraph of the
discussion section.
Response : We have added it, thank you for your review